# Causal debiasing for unknown bias in histopathology—A colon cancer use case

Ramón L. Correa-Medero[1,2], Rish Pai[3], Kingsley Ebare[3], Daniel D. Buchanan[4,5,14], Mark A. Jenkins[5,15], Amanda I. Phipps[6,9], Polly A. Newcomb[6,9], Steven Gallinger[7,12,13], Robert Grant[7,10,11], Loic Le marchand[8], Imon Banerjee[1,2] *

1 School of Computing And Augmented Intelligence Arizona State University, Phoenix, Arziona, United States of America, 2 Mayo Clinic Arizona Department of Radiology, Phoenix, Arziona, United States of America, 3 Department Of Pathology Mayo Clinic Arizona, Phoenix, Arizona, United States of America, 4 Colorectal Oncogenomics Group, Department of Clinical Pathology, The University of Melbourne, Melbourne, Victoria, Australia, 5 University of Melbourne Centre for Cancer Research, Victorian Comprehensive Cancer Centre, Melbourne, Victoria, Australia, 6 Public Health Sciences Division, Fred Hutchinson Cancer Center, Seattle, Washington, United States of America, 7 Lunenfeld-Tanenbaum Research Institute, Sinai Health System, Toronto, Ontario, Canada, 8 Department of Epidemiology, University of Hawaii, Honolulu, Hawaii, United States of America, 9 Department of Epidemiology, University of Washington, Seattle, Washington, United States of America, 10 Vector Institute, Toronto, Ontario, Canada, 11 Division of Medical Oncology and Hematology, Princess Margaret Cancer Centre, Toronto, Ontario, Canada, 12 Ontario Institute for Cancer Research, Toronto, Ontario, Canada, 13 Hepatobiliary Pancreatic Surgical Oncology Program, University Health Network, Toronto, Ontario, Canada, 14 Genomic Medicine and Family Cancer Clinic, Royal Melbourne Hospital, Parkville, Victoria, Australia, 15 Centre for Epidemiology and Biostatistics, Melbourne School of Population and Global Health, Carlton, Victoria, Australia

* Banerjee.Imon@mayo.edu

**Data Availability Statement:** The data provided by Colon Cancer Family Registry (CCFR) are available by authorized request in dbGaP. Specific details regarding the data access can be found at https://

## Abstract

Advancement of AI has opened new possibility for accurate diagnosis and prognosis using digital histopathology slides which not only saves hours of expert effort but also makes the estimation more standardized and accurate. However, preserving the AI model performance on the external sites is an extremely challenging problem in the histopathology domain which is primarily due to the difference in data acquisition and/or sampling bias. Although, AI models can also learn spurious correlation, they provide unequal performance across validation population. While it is crucial to detect and remove the bias from the AI model before the clinical application, the cause of the bias is often *unknown*. We proposed a *Causal Survival model* that can reduce the effect of unknown bias by leveraging the causal reasoning framework. We use the model to predict recurrence-free survival for the colorectal cancer patients using quantitative histopathology features from seven geographically distributed sites and achieve equalized performance compared to the baseline traditional Cox Proportional Hazards and DeepSurvival model. Through ablation study, we demonstrated benefit of novel addition of latent probability adjustment and auxiliary losses. Although detection of cause of unknown bias is unsolved, we proposed a causal debiasing solution to reduce the bias and improve the AI model generalizibility on the histopathology domain across sites. Open-source codebase for the model training can be accessed from https://github.com/ramon349/fair_survival.git

coloncfr.org/for-researchers/db-gap/. Within the scope of this study, this data was used in accordance with the terms and conditions set forth by CCFR. Due to the CCFR data sharing restrictions, we are unable to include the sample data. Researchers interested in using these data should contact CCFR directly for information on how to obtain permission. Interested person need to submit an application or a data request form along with research proposal and need to sign a Data Use Agreement (DUA) that outlines the terms and conditions under which you can use the data. For any inquiries regarding data access or if you need assistance in obtaining third-party data from CCFR, please contact Mayo Clinic study site manager Aatikah Mouti at mouti.aatikah@mayo.edu.

**Funding:** The author(s) received no specific funding for this work.

**Competing interests:** The authors have declared that no competing interests exist.

# 1 Introduction

Definitive diagnosis of tissue lesions often requires the use of histopathology analysis. Digitization of standard tissue glass slides into Whole Slide Images (WSIs) has opened a new possibility for the use of advanced computer vision techniques not only for diagnosis but also for prognosis of advanced diseases [1, 2]. Colorectal cancer is particularly challenging as highly complex image appearance often results in uncertain prognosis [3]. It thus serves as the ultimate use case for computer vision, where an artificial intelligence (AI) model can be trained to learn patterns indicative of poor prognosis. Pai et al. developed an AI-based model to reliably stratify colon cancer samples by their risk for recurrence better than using standard risk factors. [4]. Introducing potential for AI models to improve the workloads of human pathologists. Such models can even be applied to other populations where the availability of expert pathologists may be limited [2].

Deployment of AI models depends on the validation of model performance on external datasets [5]. However, the accuracy of the model is often significantly degraded on external datasets, as reported by many studies in the literature. Drops in performance are usually found to be caused by models being biased to spurious patterns in the training data [6]. As a result, many machine learning methods are unsuccessful when applied to data from unseen hospital sites despite achieving promising results on their development dataset. In addition, disparities in model predictions can reflect and thus propagate existing health disparities among underrepresented populations [7]. In a recent study, Howard et al. [8] reported that some models trained on TCGA datasets use detected source site information to predict prognosis or mutation states. Such performance could be based on the fact that the distribution of clinical information, such as survival and gene expression patterns, significantly differs among the samples provided by various laboratories. Many authors [9] considered differences in slide staining as a primary factor for the imaging bias and tried to solve it by color augmentation and stain-normalization [10–12]. In addition to acquisition-specific bias, such as scanner configuration and noise, stain variation, and artifacts, bias may exist in the targeted population. For example, Zech et.al. [13] showed that AI can diagnose pneumonia from radiology images, but it uses radiographic markers used in portable intensive care unit scanners as surrogates. Similarly, Rueckel et al. found a pneumothorax detection model that used shortcuts based on inserted chest tubes [14]. It has also been observed that imaging AI models learn spurious age, sex, and race correlations from images when trained for seemingly unrelated tasks [15]. More concerning direction, studies have shown AI imaging models demonstrate bias against historically underserved subgroups of age, sex, and race in disease diagnosis [16].

Given the complexity of data collection and the risk of associated bias, a significant amount of work has been done to solve bias using computational techniques [17]. Bias mitigation techniques focus on the explicit use of target biasing attributes to reduce disparities. However, this also serves as a limiting factor, given that the biasing variable must be known and made available for the development of a new model. Targeting one particular attribute leads to another challenge known as the "whack-a-mole" phenomenon, where other biasing attributes are then enhanced, restarting the cycle [18].

Computational approaches view generalization as two possible types of data shift. Domain shift sees the distribution of input features to models change across dataset sites. This could be due to differences in staining or different acquisition devices. Label shift, on the other hand, is a change in the labels with the same image appearance. We observed this event where different institutions/professionals disagreed on labeling a slide as either stage 2 or stage 3, subtle differences that impact model performance.

Alabdoulmohsin et al. proposed that the key challenge to generalization across sites is a mixture of domain and label shift, making the use of only one technique suboptimal. This work discusses how the dual domain and label shift is caused by a change in the latent population. This latent population is present across all data sites; however, its occurrence across domains is shifted such that the relationship between patients and diseases is changed. The main challenge is thus to identify this latent population in order to leverage existing adaptation techniques. Their group introduced a causal analysis framework where this hidden population captured by hidden variable $U$ can be learned from the data should several additional proxy variables be available. The causal diagram in Fig 3 demonstrates the relationship between the hidden variable $U$, the task variable $Y$, and the image feature $X$. Two additional variables, proxy $W$ and $C$, are introduced to better estimate the hidden variable $U$. The variable $C$ is a concept variable following the work of Koh et al. [19]. Concepts are used to guide a model in learning feature representations directly related to the main task through auxiliary training tasks. The proxy variable $W$, identified by researchers, provides a potential glimpse into the hidden population in the data and is used to guide the learning of the latent population. The group proposed a methodology to learn the hidden population indicator and used domain adaptation techniques to adapt the model to new populations, lessening the effect of the shift on the model performance.

In this work, we propose a novel survival model by incorporating the concept of latent-shift to reduce the effect of unknown bias by treating the generalization as a domain adaptation problem. We followed Alabdulmohsin et al. [20] unsupervised domain adaptation technique to the latent distribution shifts, which generalize the standard settings of covariate and label shift of the domain. We leverage auxiliary data in the source domain in the form of tumor staging and hospital site information, a proxy for socioeconomic status, and apply it to derive an identification strategy for the optimal predictor under the target distribution.

## 2 Methods

### 2.1 Dataset

The study population consisted of patients with colorectal carcinoma from the Colon Cancer Family Registry (CCFR, participating sites: Seattle, Australia, Mayo Clinic, Ontario, and Hawaii) as well as three sites external to the CCFR: University of Pittsburgh Medical Center (UPMC), Mount Sinai Hospital Toronto, and Seattle-Puget Sound (Access) Cancer Registry [21]. The CCFR enrolled participants after colorectal carcinoma diagnosis with prospective follow-up. The UPMC and Mount Sinai cohorts consisted of consecutively resected colorectal carcinoma at these institutions between 2010 and 2015 and 2011 and 2016, respectively. The Seattle-Puget Sound Cancer Registry cohort has been previously described and consists of patients between 20 and 74 years of age diagnosed with CRC between 2016 and 2018. Recurrence was assessed by manual review of medical records and was available for 3,349 stage I–III CRCs with a median follow-up of 58 months. For the prognostic model training and external validation, the stage I–III CCFR CRCs with recurrence data (n = 2,411) formed the internal cohort, and the UPMC, Mount Sinai, and Seattle-Puget CRCs (n = 938) formed the external validation cohort. This study was approved by the Mayo Clinic institutional review board (IRB 806–96 and 18–11309). The data was available to the research team on Jan 14, 2019. No patients were involved in any part of the study, including concept and study design, data collection, analysis and interpretation, drafting of the manuscript, and critical revision.

Table 1 shows the center, disease, and demographic characteristics of train, validation, and test sets that were randomly selected. Fig 1 shows the survival data for both internal and external sites Difference in survival rates between populations was measured using the log-rank

**Table 1. Description of the dataset—Shows the characteristics of both internal and external datasets.**

|  |  | Grouped by Data Split | | | | |
|---|---|---|---|---|---|---|
|  |  | **Overall** | **external-test** | **internal-test** | **internal-train** | **internal-val** |
| n | – | 3411 | 946 | 766 | 1456 | 243 |
| center, n (%) | ACCESS | 128 (3.8) | 128 (13.5) | – | – |  |
|  | Mt. Sinai | 361 (10.6) | 361 (38.2) | – | – |  |
|  | UPMC | 457 (13.4) | 457 (48.3) | – | – |  |
|  | Australia | 233 (6.8) | – | 75 (9.8) | 136 (9.3) | 22 (9.1) |
|  | Hawaii | 37 (1.1) | – | 8 (1.0) | 25 (1.7) | 4 (1.6) |
|  | Mayo | 561 (16.4) | – | 182 (23.8) | 326 (22.4) | 53 (21.8) |
|  | Ontario | 1118 (32.8) | – | 347 (45.3) | 657 (45.1) | 114 (46.9) |
|  | Seattle | 516 (15.1) | – | 154 (20.1) | 312 (21.4) | 50 (20.6) |
| Stage, n (%) | 1 | 689 (20.2) | 209 (22.1) | 151 (19.7) | 278 (19.1) | 51 (21.0) |
|  | 2 | 1364 (40.0) | 376 (39.7) | 292 (38.1) | 601 (41.3) | 95 (39.1) |
|  | 3 | 1358 (39.8) | 361 (38.2) | 323 (42.2) | 577 (39.6) | 97 (39.9) |
| Time to event, mean (SD) |  | 42.2 (20.7) | 38.3 (22.2) | 43.9 (19.9) | 43.8 (19.9) | 42.5 (20.3) |

test, a metric which quantifies the difference in survival rates between two populations [22]. P-values from the log-rank test are measured by considering Ontario, the largest site, as a reference. We found survival rates from Mayo Clinic and UPMC are not significantly different from the Ontario data. However, the remaining sites, Seattle, Australia, Mt.Sinai, and Access, have significantly different survival rates (based on p-values).

## 2.2 Quantitative image feature extraction

We followed the standard histopathology image acquisition pipeline for image staining and scanning; however, given the geographical distance, the slides were scanned at different

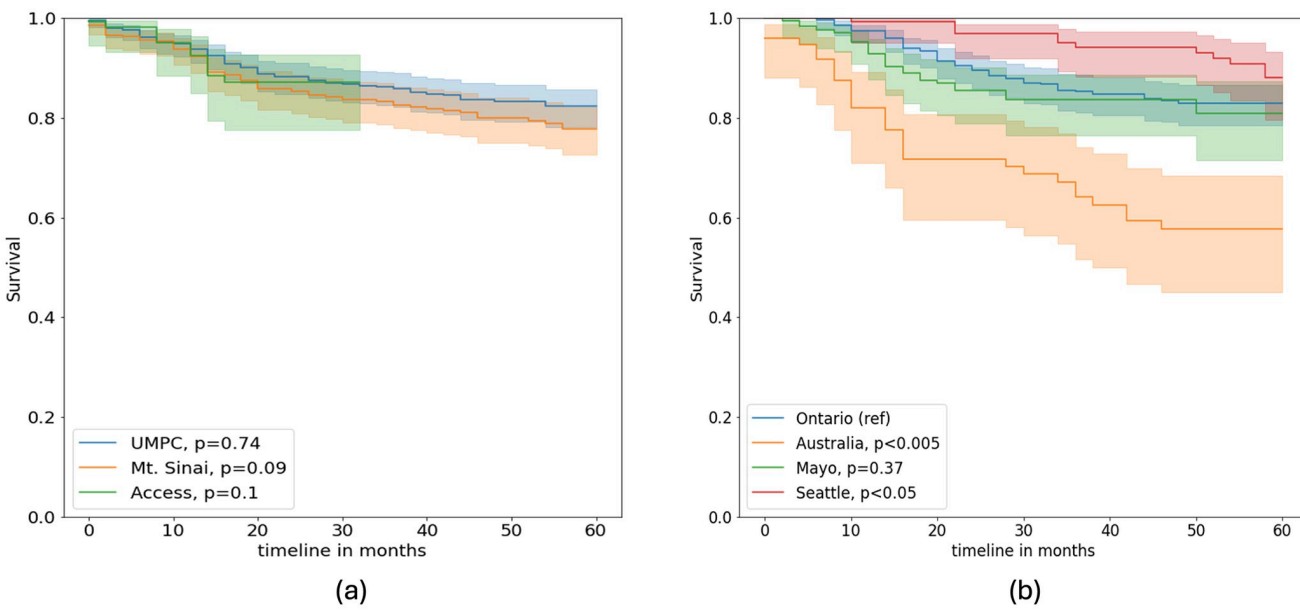

**Fig 1. Recurrence-free survival for the colorectal carcinoma patients—(a) internal and (b) external sites.** P-values are computed as a log-rank test using the internal site Ontario as a reference (ref).

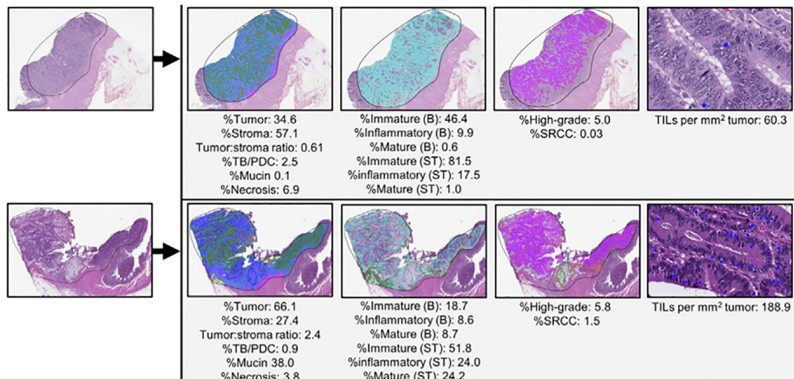

**Fig 2. Quantitative feature extraction pipeline on 2 sample images, which segments the image in a stepwise manner.** First, the image is segmented into carcinoma (green), stroma (light blue), mucin (dark blue), TB/PDC (red), necrosis (brown), smooth muscle (purple, and fat (yellow). Next, the stroma is segmented into immature (teal), mature (green), and inflammatory (gray). The carcinoma is segmented into low-grade (purple), high-grade (orange), and signet ring cells (light green). Finally, TILs are recognized as objects (blue dots) within the tumor. After this segmentation, 15 features are calculated from each image as shown. Abbreviations: B, tumor bed; ST, stromal region.

locations. From each of the surgically resected colorectal cancer studies, one representative H&E slide was digitized using Leica Aperio GT450 or AT2 at ×40 magnification. The images from the internal cohort were stained with H&E and scanned at Mayo Clinic. The Seattle-Puget cases were stained with H&E at Fred Hutchison Cancer Center and scanned at Mayo Clinic. The UPMC and Mount Sinai cases were stained with H&E and scanned at their respective institutions. After staining, all images were uploaded to the Aiforia Create deep learning cloud-based platform (Aiforia Technologies, Helsinki, Finland), a commercially available platform designed explicitly for histologic images. Each image was manually reviewed by an expert gastrointestinal pathologist (Dr. Pai), and the entire tumor bed was outlined (median 96.5 $mm^2$ analyzed per CRC).

Previously developed deep learning quantitative segmentation algorithm, QuantCRC [23], was applied to the tumor bed to segment colorectal carcinoma digitized images into 13 regions and one object (Fig 2). The QuantCRC algorithm uses convolutional neural networks (CNN) to segment the image in a stepwise manner and is trained using 24,157 annotations made on 559 images, which are not used in this study. First, the tumor bed is segmented into carcinoma, TB/PDC, stroma, mucin, necrosis, fat, and smooth muscle. The second layer segments stroma into immature (loose, often myxoid stroma with haphazardly arranged plump fibroblasts and collagen fibers), mature (densely collagenous areas with scattered fibroblasts, often with parallel collagen fibers), and inflammatory (dense clusters of chronic inflammatory cells obscuring stromal cells) subtypes. The third layer segments carcinoma into low-grade, high-grade, and signet ring cell carcinoma (SRCC). The fourth layer identifies TILs. To re-validate QuantCRC, 30 images (15 from each GT450 and AT2 scanners) were selected from the 3,349 CRCs with recurrence data. For layers 1 to 3, the algorithm output was compared with annotations by five gastrointestinal pathologists. The results from the segmentation algorithm compare favorably with annotations by gastrointestinal pathologists for all features. The most variation was seen between immature and mature stroma, where disagreement among the five pathologist raters was noted. This is likely related to the fact that stroma subtyping is a new concept and is not done routinely by pathologists for clinical practice.

For each image, fat and smooth muscle were subtracted from the tissue area to generate the tumor bed area. From the tumor bed area, the following 15 quantitative parameters were

measured: %tumor, %stroma, tumor:stroma ratio, %TB/PDC within the tumor, %mucin within the tumor, %necrosis within the tumor bed, %high-grade, %SRCC, TILs per mm2 of tumor, %immature stroma (tumor bed), %inflammatory stroma (tumor bed), % mature stroma (tumor bed), %immature (stromal region), % inflammatory (stromal region), and % mature (stromal region). These 15 quantitative parameters comprise the QuantCRC features used for downstream analysis.

## 2.3 Causal survival model

Fig 3 presents the proposed causal modeling framework for recurrence-free survival estimation across multiple sites. Our primary hypothesis is that by using the proxy ($W$) and auxiliary ($C$) variable in a causal framework, we will be able to adapt to the latent subgroup shift that appears between sites without knowing the cause of bias and able to predict the survival $Y$ given the input $X$ (Fig 3.i). Following [20], if we have two domains ($P$ source and $Q$ target), we frame the learning $Q(Y|X)$ as identification problem where observations are drawn from $P(X, C, Y, W)$ and $Q(X)$. We assume that $C$ and $W$ are observed in the source distribution and thus can be used during learning when $U$ is unobserved and $U$ is a discrete variable. We also assume that conditional dependencies in the data exist iff they exist in the graph (Fig 3.i). Note that we only observe ($X, C, Y, W$) in the source $P$ and $X$ in the target $Q$.

We adopted the latent shift causal framework for predicting recurrence-free survival ($Y$) of colorectal carcinoma patients using the quantitative histopathology features ($X$). Given the issue of not knowing the unobserved bias variable, we model the treatment site as a proxy ($W$) and the cancer stage as a concept ($C$). We assume that input histopathology features ($X$) are affected by the site due to the bias in acquisition and population variations, and the cancer stage directly affects recurrence-free survival ($Y$). The learning uses three core components (Fig 3).

**2.3.1 Auto encoder training (Module A).** First, in module A, we learn the discrete unobserved latent variable $U$ by training an auto encoder to reconstruct the input, i.e., QuantCRC

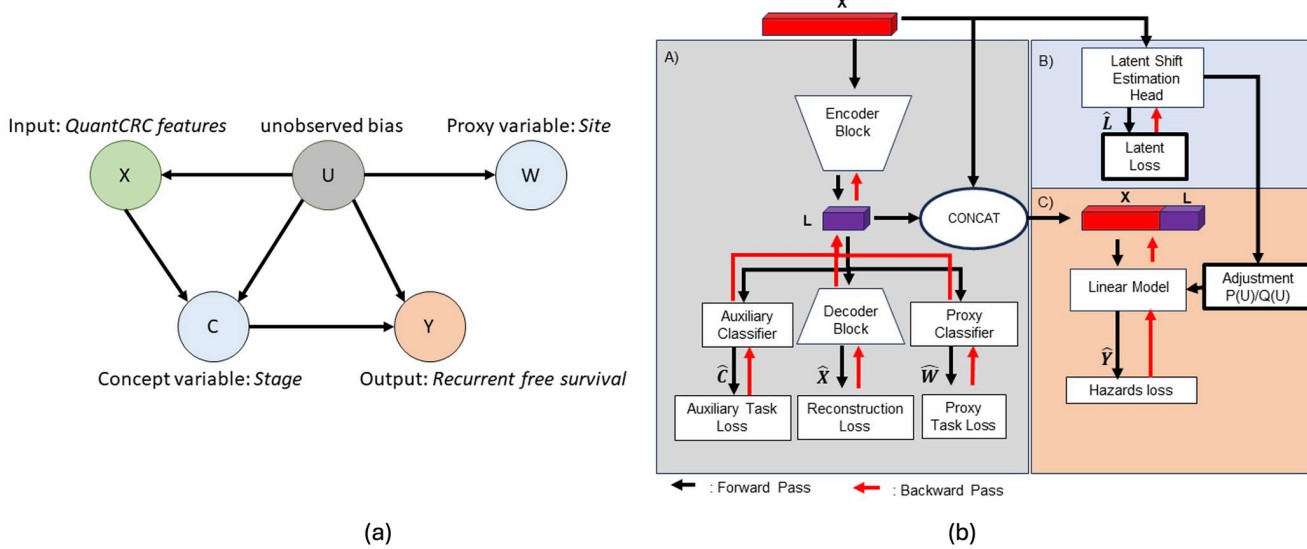

(a)                     (b)

**Fig 3. Causal modeling framework: (left) latent shift assumption and causal relations; (right) proposed model diagram demonstrates 3 main components of the model.** A) learns a latent representation capturing task relevant information alongside proxy information. B) Attempts to infer the latent variables directly from the input features. C) A risk estimation model is trained, predictions are modified using the latent estimates.

features. The auto-encoder consists of a single linear layer that produces a smaller ten-dimensional tensor. Followed by a single decoder layer that reproduced the original input. Mean squared error loss is used to guide the auto-encoder's training. The latent space is constrained to make a discrete representation using the Gumbel-softmax trick where through annealing the temperature value, the output probabilities become discrete one hot representation [24]. Allowing us to ultimately use the latent space to represent the discrete latent variable $U$. In addition to the reconstruction loss for the features, the encoder component is also trained with two auxiliary classifiers—(i) classify tumor staging (classify concept vector $C$), (ii) predict hospital of origin (predict the proxy $W$). The classification task uses categorical cross-entropy losses. KL divergence ($KLD$) against uniform prior $\tilde{U}$ is added to reduce the potential for prior collapse, a known issue in autoencoders. Eq 1, demonstrates the final loss for training the auto-encoder model where $X$ is the input and reconstructed variables are represented with a hat.

$$L_{total} = \beta_X L_{MSE}(X, \hat{X}) + \beta_C L_{CE}(C, \hat{C}) + \beta_W L_{CE}(W, \hat{W}) + \beta_{kl} L_{KLD}(\hat{U}, \tilde{U}) \qquad (1)$$

**2.3.2 Latent estimation (Module B).** To properly estimate the latent shift between source $P$ and target site $Q$, we would need a reliable estimate of the latent variable probabilities. Fig 3 module B demonstrates the parallel latent estimation branch trained to predict $U$ from the original QuantCRC features directly without referring to the proxy and concept loss. The auto-encoder is frozen and provides the ground truth latent variable labels. The estimation branch is thus trained using the binary cross entropy loss between predicted latent $\hat{U}$ and auto encoder latent $U$. The outputs of this module are used to modify the final risk prediction seen in Eq 2 by providing the terms $P(U = i|X)$, $Q(U = i)$, $P(U = i)$.

## 2.4 Recurrence risk prediction (Module C)

In the final stage of the training process (module C), we train a final risk prediction model that leverages the QuantCRC features ($X$) and learned latent variable $U$ (module A) to predict the risk of recurrence at different time points. The model is trained using the negative log-likelihood loss until convergence. Adaptation to the target domain is done using the QuantCRC features of the external data (domain Q), the latent estimation branch (module B) is used to predict latent variables, and a shift ratio is then estimated. The estimates from the training data (domain $P$) and external data (domain $Q$) are compared, and the difference is used to re-scale the model predictions (see Eq 2). The recurrence risk estimation head (module C) is then fine-tuned using the internal data ($P$) to account for the new latent shift estimations based on an estimate of $U$.

$$HR_Q(t, X) = H_0(t) \sum_{i=0}^{k_u} HR_P(X, U = i) P(U = i|X) \frac{Q(U = i)}{P(U = i)} \qquad (2)$$

During inference, the model uses the predicted latent shift of the new data to re-scale predictions. For our use case, we will treat samples from the external dataset ($Q$) as a single domain for which we adjust our model predictions. All models were trained using the same parameters: batch size of 256, Adam optimizer with a learning rate of 0.01, and early stopping based on validation loss.

## 2.5 Model evaluation

Model performance was evaluated for two essential metrics. We first leveraged the concordance index (C-index) to evaluate the predictive power of our mode [25]. The C-index is

similar to the area under the receiver operating curve metric with the extension of being aware of time and occurrence of events of interest. The metric functions by comparing concordant and discordant pairs based on the ordering of events and the individual's risk factor. The second metric we used was to evaluate the AUC over time, where for each time point in the study, 2- 58 months, we evaluated the predictive power of the model's risk score to predict recurrence. The main distinction is that the AUC is evaluated across each time point. At the same time, the C-index will provide a single value for the model's predictions while accounting for the occurrence of censoring. Furthermore, to provide a robust statistical comparison, we ran a procedure known as auto-bootstrapping. Where the test set is subsampled randomly over 100 iterations, and performance is calculated on a randomly selected subset. A 95% confidence interval is calculated for our concordance metric to measure the robustness. Each bootstrap run utilized the same random seed for reproducibility and fair comparison, ensuring each model was evaluated on the same subset of samples.

## 3 Results

Feature correlation was measured using the Pearson correlation coefficient to find a link between center and survival as a potential source of bias. Fig 4 shows the pair-wise Pearson correlation coefficient values between QuantCRC features, stage, center, and recurrence. As we can observe, individual features do not have a high correlation ($>0.5$) with recurrence ($Y$);

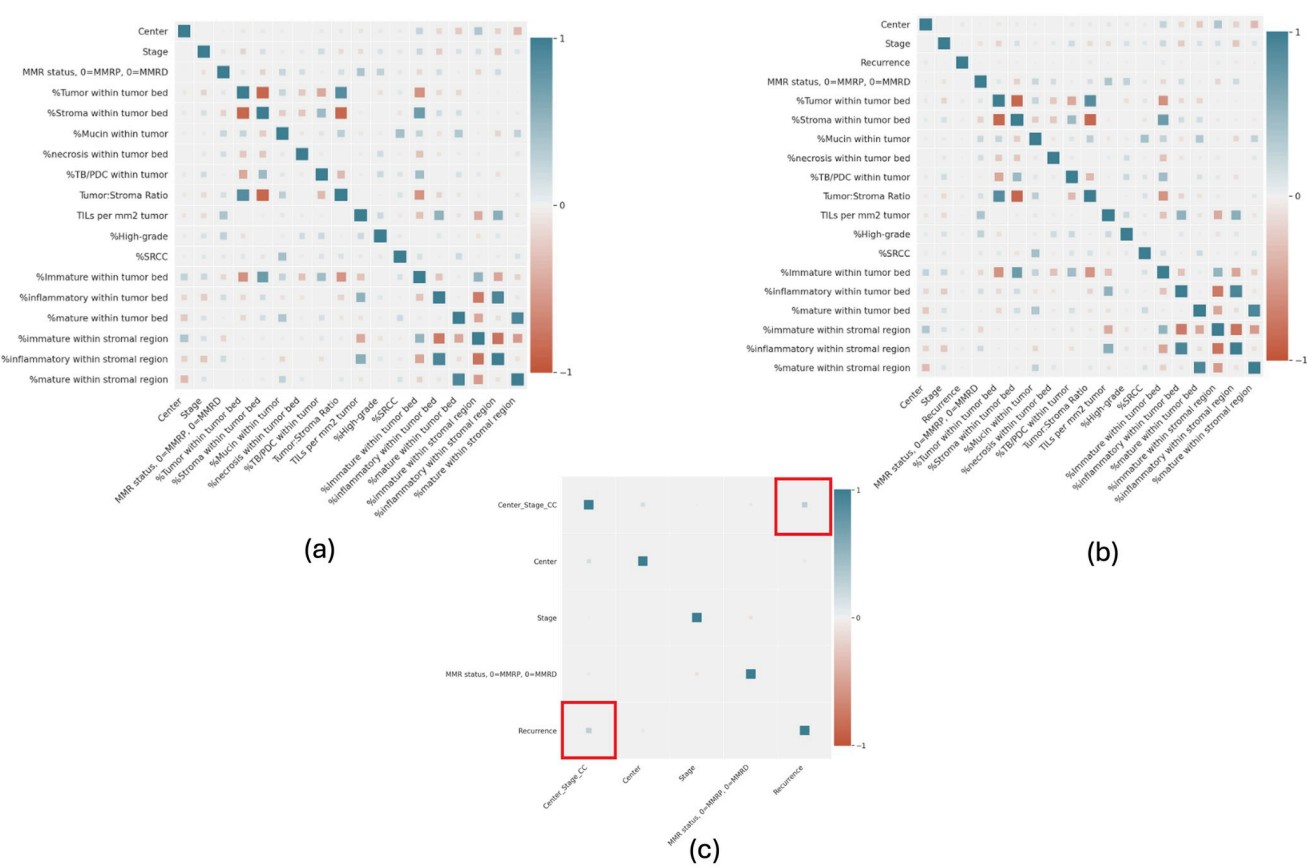

**Fig 4. Pearson correlation coefficient between Stage ($C$), Center ($W$), QuantCRC features ($X$) and target recurrence ($Y$)—(a) represents stage and center as separate variables, (b) represents stage and center as a combined variables, and (c) Correlation with recurrence.**

**Table 2. Performance of the models on the internal and external test sets in-terms of C-index.** 95% confidence interval is calculated using auto-bootstraping. **Bold** front represents optimal performance.

| Data | Center | Baseline(Cox Model) | DeepSurv | Causal Survival |
|---|---|---|---|---|
| Internal | Australia | 0.644 (0.601,0.686) | 0.641 (0.598,0.684) | **0.673 (0.636,0.71)** |
| | Mayo | 0.703 (0.651,0.754) | 0.734 (0.686,0.782) | **0.736 (0.682,0.79)** |
| | Ontario | **0.658 (0.630,0.685)** | 0.609 (0.582,0.637) | 0.640 (0.611,0.67) |
| | Seattle | 0.727 (0.648,0.807) | **0.775 (0.704,0.847)** | 0.737 (0.660,0.813) |
| External | ACCESS | 0.643 (0.550,0.736) | 0.632 (0.532,0.732) | **0.660 (0.571,0.749)** |
| | Mt. Sinai | **0.698 (0.666,0.73)** | 0.661 (0.629,0.693) | 0.684 (0.653,0.714) |
| | UPMC | 0.711 (0.689,0.734) | 0.675 (0.648,0.703) | **0.716 (0.692,0.739)** |
| Overall Internal | | 0.696 (0.677,0.714) | 0.687 (0.665,0.708) | **0.702 (0.683,0.72)** |
| Overall External | | 0.698 (0.675,0.721) | 0.651 (0.626,0.676) | **0.699 (0.675,0.723)** |

however, stroma and tumor bed-related features are highly correlated among themselves and also moderately correlated with center and stage. Interestingly, a combination of center and stage increase the correlation with the target recurrence $Y$, which follows our hypothesis that the unobserved variable ($U$) that affects the outcome ($Y$) can be estimated from the stage ($C$) and center ($W$).

As a baseline, we compared our proposed model performance against the traditional Cox Proportional Hazard (CPH) model trained using the lifelines package [26] and DeepSurv model [27]—a multi-layer deep neural network with hazard loss. Model evaluation was done by measuring the concordance index of the model's risk score at 60 months, comparing the ranking of at-risk patients adequately. The 95% confidence interval was obtained for all the model performances using the autobootstraping. Overall and site-based performance is presented in Table 2.

Overall, the CPH baseline (CI internal: 0.696, external:0.698) and Causal Survival (CI internal:0.702, external:0.699) models achieve comparable performance on the overall internal and external test sets while DeepSurv performance is consistently lower with higher variability. However, the performance differences between the sites are significant in the baseline Cox model, e.g., the C-index for Seattle is 0.727, and for Australia is 0.644. While maintaining the overall performance, the Causal survival model reduces the disparities between the site performance, e.g., the C-index for Seattle is 0.737, and for Australia is 0.673. Similar results can be observed for the external dataset, such as that the causal survival model improved the performance of all the sites, including ACCESS.

In Fig 5, we represent comparative performance between the models in terms of the area under the operating characteristics curve (AUROC) at different time intervals. Causal survival model performance is consistently better for all the time points on the internal dataset, except two months, which has the lowest rate of recurrence. A similar consistent performance boost was observed for the external data for predicting short-term as well as long-term recurrence. On the external data, causal survival model has a huge performance boost over DeepSurv ($\approx$ 5% AUC).

### 3.1 Ablation

To demonstrate the efficacy of our proposed adaptation of the causal domain adaptation framework, the modularized framework allows us to evaluate the effect of key modules on the final predictions. Table 3 highlights the ablation performance for different configurations.

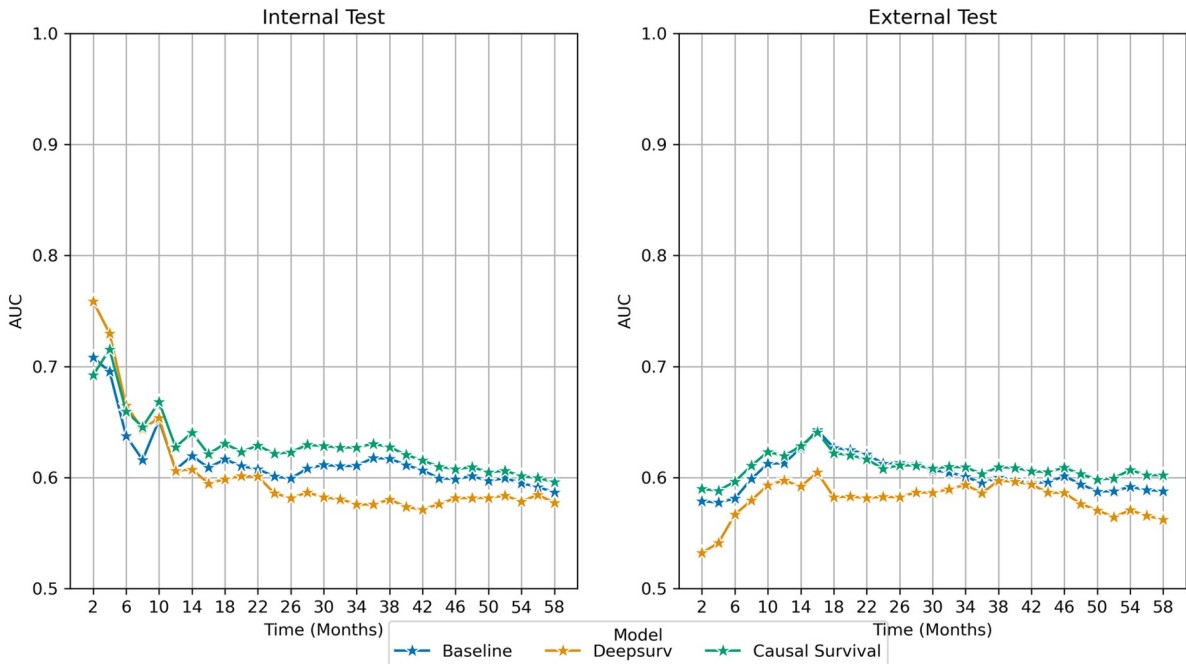

**Fig 5. Comparative area under the receiver operating characteristics curve (AUROC) calculated for every time step on the internal and external dataset—(left) internal and (right) external.** Blue: Baseline Cox model, Orange: DeepSurv, and Green: Proposed Causal Survival model.

i. *Removal of Latent Shift Adjustment (dropped module C)*: First, we remove the proposed domain adjustment component from module C, i.e. $P(U)/Q(U)$ and evaluate the performance on the internal and external test data. As expected, the removal of latent shift domain adjustment actually improved the performance on the internal data ($P$) but dropped the performance on the external test ($Q$). This shows the fact that the proposed latent shift adjustment helps to improve the generalizability of the model on the external data.

ii. *Infer only using Latent Variable (dropped module B and C)*: As the second experiment, we only use the latent variable derived by module A and pass it through hazard layer to compute the survival risk. The performance of the model remains high in the internal test, which demonstrates the latent features from the encoder are able to capture task-relevant correlations. Given the latent estimation head and the adjustment parameter for the domain $P$ are missing, the performance dropped on the external datasets.

iii. *Infer only using QuantCRC features (dropped modules A, B, and C)*: In the final experiment, we only used the QuantCRC features. We observed that the model performance decreased

**Table 3. Configuration of ablation study.** Removed components of the causal survival model is marked as X. Performance measured as C-index with 95% confidence interval.

| Latent Shift Probability Adjustment | Infer Using Latent Variable | Infer using Quant CRC Features | Internal-Test Concordance Index | External Test Concordance Index |
|---|---|---|---|---|
| X | | | 0.712 (0.694,0.73) | 0.691 (0.669,0.713) |
| | X | | 0.538 (0.513,0.562) | 0.486 (0.462,0.51) |
| | | X | 0.702 (0.683,0.721) | 0.686 (0.661,0.711) |

to be in range with the Deepsurv model in the internal test set. Performance on the external set Drops to the lowest amongst all models. Suggesting the predictive power of the latent features adjusted based on proxy and concept is essential for proper risk prediction.

## 4 Discussion

Bias in AI models for the histopathology domain can arise from several sources—(i) *Data bias*: If the training data is not representative of the whole population, the model may not generalize well on new data. For example, if the training data only includes samples from one scanner, the model may not perform well on samples from other acquisition devices; (ii) *Algorithmic bias*: The algorithms used to train the model may introduce bias. For example, if the algorithm is designed to optimize for a specific metric (e.g. AUC), it may ignore other important factors. Such as disparities between the ethnic subgroups; (iii) *Human bias*: Human bias can be introduced during the data annotation process or when selecting the data used to train the model. It is important to identify and address these sources of bias to ensure that the AI model performs accurately and fairly. There existing supervised techniques in the field of domain adaptation that attempt to mitigate biases that are known. However, the true cause of bias is unknown or related to multiple factors.

In this work, we proposed a causal survival model that can reduce the effect of unknown bias via a causal reasoning framework incorporated within a deep learning paradigm. As our first use case, we evaluated the model for predicting recurrence for colorectal cancer patients across seven geographically distributed sites using the Colon Cancer Family Registry (CCFR) and showed improvement in performance across sites. Our primary contribution is to adapt the unsupervised domain adaptation technique to adjust the deep latent space of the out-of-domain samples and ultimately obtain analogous performance across in-domain and out-of-domain samples.

Interestingly, as shown in the correlation plots (Fig 4), cancer stage and recurrence-free survival across sites have limited to no correlation. Indicating that other factors beyond staging can mediate the risk of recurrence for colorectal cancer patients. Our causal modeling approach suggests that the models are able to learn to measure the hidden variations in data, in other words, 'unknown bias', and utilize it to improve prediction generalizability and efficiently handle distribution shifts between internal and external data. In particular, we observe that survival rates from Australia, Seattle, Mt. Sinai, and Access are significantly different ($p < 0.1$) from the reference internal site Ontario. Additionally, there is a positive correlation between the linear combination of center and stage and recurrence-free survival. The disparity is reflected in the models' performance, where the baseline models (Cox and DeepSurv) underperformed or overperformed on these sets. Our proposed architecture demonstrates an overall improvement in both internal and external datasets. We observe a significant improvement in the risk prediction on the three external datasets without utilizing fine-tuning or data harmonization techniques. Our ablation study demonstrated that by learning the latent features with auxiliary losses based on potential bias factors, an improvement in model predictions can be obtained. Finally, incorporating the latent shift adjustment further guarantees the stability of model predictions in a new domain.

The proposed framework is an adaption of the unsupervised latent distribution shift method, where we introduce deep latent space features and a domain adjustment factor. We also extend the framework for the prognosis of recurrence-free survival for colorectal cancer patients using quantitative histopathology features. Though detection and interpretation of 'unknown bias' is still an unsolved challenge, our proposed solution estimates the bias via the

proxy and concept variable and reduces the bias to improve AI model generalizability. This framework could be extended to other applications, such as breast cancer risk prediction and chest x-ray pathology detection, where it is possible to incur both domain and label shifts. The methodology could assist research in moving from a single source of the biasing attributes to tackle the true distribution shift by using the population latent shift.

## 4.1 Limitation

The study has several limitations. First, the causal survival framework requires the availability of two mediating concepts $C$ and of a proxy variable $W$ at training time. These measures might not be readily available for other studies, or they may not satisfy all the assumptions described in the framework. Another limitation is the quality of the auxiliary variables. In other studies, the variables may not be as easily learnable, producing an unreliable network. Furthermore, the causal assumptions are typically not testable as $U$ is not observed. Second, we validated the model on the quantified histopathology features, but ideally, the paradigm can also be used on the raw image data. Third, we only observed a moderate improvement in performance due to the fact that the slides are scanned and digitized using a standard protocol, reducing variations between the scanned slides within CCFR. Furthermore, manual annotation variations were reduced in our data by having a single expert gastrointestinal pathologist drawing the tumor boundary. We believe that a scenario where multiple sites contribute to clinically scanned slides without a common protocol will result in site-specific variations amplifying biases. In such a case, our proposed method will provide better performance over baseline survival models.

## Supporting information

**S1 Data.**
(ZIP)

## Author Contributions

**Conceptualization:** Rish Pai, Imon Banerjee.

**Data curation:** Rish Pai, Kingsley Ebare, Daniel D. Buchanan, Mark A. Jenkins, Amanda I. Phipps, Polly A. Newcomb, Steven Gallinger, Robert Grant, Loic Le marchand.

**Formal analysis:** Ramón L. Correa-Medero, Imon Banerjee.

**Investigation:** Imon Banerjee.

**Methodology:** Ramón L. Correa-Medero, Imon Banerjee.

**Project administration:** Imon Banerjee.

**Resources:** Rish Pai, Imon Banerjee.

**Software:** Ramón L. Correa-Medero, Imon Banerjee.

**Supervision:** Ramón L. Correa-Medero, Rish Pai, Imon Banerjee.

**Validation:** Ramón L. Correa-Medero, Imon Banerjee.

**Visualization:** Ramón L. Correa-Medero.

**Writing – original draft:** Ramón L. Correa-Medero, Imon Banerjee.

**Writing – review & editing:** Ramón L. Correa-Medero, Rish Pai, Kingsley Ebare, Daniel D. Buchanan, Mark A. Jenkins, Amanda I. Phipps, Polly A. Newcomb, Steven Gallinger, Robert Grant, Loic Le marchand, Imon Banerjee.

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
