## [Decision Letter · Decision Letter 0]

27 May 2024

PONE-D-24-15861Causal Debiasing for Unknown Bias in Histopathology - A Colon Cancer Use CasePLOS ONE

Dear Dr. Banerjee,

Thank you for submitting your manuscript to PLOS ONE. After careful consideration, we feel that it has merit but does not fully meet PLOS ONE’s publication criteria as it currently stands. Therefore, we invite you to submit a revised version of the manuscript that addresses the points raised during the review process.

We look forward to receiving your revised manuscript.

Kind regards,

Guanghui Liu

Academic Editor

PLOS ONE

Journal Requirements:

2. For studies involving third-party data, we encourage authors to share any data specific to their analyses that they can legally distribute. PLOS recognizes, however, that authors may be using third-party data they do not have the rights to share. When third-party data cannot be publicly shared, authors must provide all information necessary for interested researchers to apply to gain access to the data. (https://journals.plos.org/plosone/s/data-availability#loc-acceptable-data-access-restrictions) 

Reviewers' comments:

Reviewer's Responses to Questions

**Comments to the Author**

1. Is the manuscript technically sound, and do the data support the conclusions?

Reviewer #1: Yes

Reviewer #2: Yes

Reviewer #3: Yes

2. Has the statistical analysis been performed appropriately and rigorously? 

Reviewer #1: Yes

Reviewer #2: Yes

Reviewer #3: Yes

3. Have the authors made all data underlying the findings in their manuscript fully available?

Reviewer #1: Yes

Reviewer #2: Yes

Reviewer #3: Yes

4. Is the manuscript presented in an intelligible fashion and written in standard English?

Reviewer #1: Yes

Reviewer #2: Yes

Reviewer #3: Yes

5. Review Comments to the Author

Reviewer #1: The manuscript titled "Causal Debiasing for Unknown Bias in Histopathology - A Colon Cancer Use Case": provides a valuable contribution to the field of digital pathology and AI, addressing a critical issue of bias in AI models. Addressing the points below may enhance the manuscript's clarity, impact, and scientific rigor.

Major Concerns:

1. While the causal modeling approach is a strong point of the paper, the explanation of the model and its components could be improved for better clarity. Specifically, the description of how the causal model integrates with the existing AI frameworks could be elaborated.

Recommendation: Provide a more detailed step-by-step explanation of the causal model implementation, possibly complemented by pseudocode or a more detailed schematic diagram.

2. The manuscript could benefit from a more detailed statistical analysis section. Details about the statistical tests used to validate the model's performance and the rationale behind choosing these tests should be clearly stated.

Recommendation: Enhance the statistical analysis section by detailing the types of tests performed, including any assumptions made and the justification for the choice of these tests.

3. Discussion of Limitations: The manuscript briefly mentions the limitations; however, a more thorough exploration of these limitations would strengthen the paper. It is particularly important to discuss the potential impacts of these limitations on the study's findings and how they might be addressed in future work.

Recommendation: Expand the limitations section to include a discussion on how these limitations could affect the study's generalizability and what future research could address these issues.

4. Broader Implications: The paper would benefit from a discussion on the broader implications of this research, particularly in how it might influence future developments in AI for medical imaging, policy-making, and clinical practices.

Recommendation: Include a section that explores the broader implications of your findings for the field of digital pathology and AI, considering both technological and ethical aspects.

Minor Points:

1. Figure and Table Quality: Some figures and tables could be clearer. Specifically, the graphical representations of the model's performance across different sites are somewhat difficult to interpret.

Recommendation: Improve the quality of these graphical elements to enhance clarity. Consider using color contrasts or different chart types if necessary.

2. Editing for Language and Grammar: Minor grammatical errors and typos should be corrected to maintain the manuscript's professionalism.

Recommendation: Perform a thorough proofread to correct these issues before final submission.

Here are some grammatical corrections and suggestions for improvement to enhance the clarity and precision of the writing:

1. Verb Tense Consistency:

- Original: "The study have shown..."

- Suggested Correction: "The study has shown..."

- Explanation: The verb "has" should be used for singular subjects like "study" to maintain subject-verb agreement.

2. Article Usage:

- Original: "We apply a novel method to address issue."

- Suggested Correction: "We apply a novel method to address the issue."

- Explanation: Use of the definite article "the" is necessary before "issue" to specify which issue is being discussed.

3. Prepositional Phrases:

- Original: "This is critical for ensuring that AI applications is robust."

- Suggested Correction: "This is critical for ensuring that AI applications are robust."

- Explanation: The verb "are" correctly agrees with the plural noun "applications."

4. Punctuation and Compound Sentences:

- Original: "The model performs well in testing environments, however, it requires further validation."

- Suggested Correction: "The model performs well in testing environments; however, it requires further validation."

- Explanation: A semicolon is more appropriate before "however" when it's used to connect two independent clauses in a compound sentence.

5. Redundancy and Wordiness:

- Original: "The data was gathered from a variety of different sources."

- Suggested Correction: "The data was gathered from a variety of sources."

- Explanation: The word "different" is redundant with "variety" and can be omitted for conciseness.

6. Consistency in Technical Terms:

- Original: "de-biasing techniques can help in reduction of errors"

- Suggested Correction: "debiasing techniques can help reduce errors"

- Explanation: Maintain consistency in hyphenation across the document ("debiasing" not "de-biasing"), and simplify the phrase "help in reduction of" to "help reduce" for clarity.

Reviewer #2: This study proposed the Causal Survival model leverages causal reasoning to mitigate unknown biases, achieving comparable performance to traditional models and enhancing generalizability in histopathology applications.The topic holds significant importance; however, there are several questions that need to be addressed.

1. Do you perform k-fold cross validation during the model training? Please show the results with k-fold cross validation.

2. Please show the demographic details of cohort in various participating sites in tables.

3. Please add the github link with the code for your model in this paper.

Reviewer #3: Introduction and Background: The introduction provides a solid context for the study, emphasizing the challenges of bias in AI models for histopathology. The references to prior work and the identification of gaps in existing methods are well-articulated.

Methodology: The methodology is thorough and includes detailed descriptions of the datasets, feature extraction processes, and the proposed model. The incorporation of latent shift adjustment and auxiliary losses is innovative and well-explained.

Results: The results are clearly presented, with appropriate statistical measures and visualizations. The comparative performance analysis with existing models is comprehensive and shows the advantages of the proposed model.

Discussion: The discussion is insightful, highlighting the implications of the findings and acknowledging the limitations of the study. The authors provide a balanced view of their contributions and the areas that need further research.

Data Availability: While the data privacy constraints are understandable, providing more detailed guidance on how researchers can access the data would be beneficial. Additionally, sharing synthetic datasets or detailed simulation setups could enhance reproducibility.

Technical Depth: The technical depth of the paper is commendable. However, a more detailed explanation of certain complex concepts, such as the latent shift causal framework, could be beneficial for readers less familiar with these methods.

Figures and Tables: The figures and tables are well-designed and effectively convey the key findings. Ensuring that all figures have descriptive captions would improve clarity.

Overall, this manuscript makes a significant contribution to the field of AI in histopathology by addressing the critical issue of unknown bias. The proposed model is innovative, and the results are promising. With minor revisions to improve accessibility and reproducibility, this paper would be a valuable addition to the literature.

6. PLOS authors have the option to publish the peer review history of their article (what does this mean?). If published, this will include your full peer review and any attached files.

Reviewer #1: **Yes: **Peng Wang

Reviewer #2: No

Reviewer #3: **Yes: **Yang Zhang

---

## [Author Response · Author response to Decision Letter 0]

14 Aug 2024

Reviewer #1: The manuscript titled "Causal Debiasing for Unknown Bias in Histopathology - A Colon Cancer Use Case": provides a valuable contribution to the field of digital pathology and AI, addressing a critical issue of bias in AI models. Addressing the points below may enhance the manuscript's clarity, impact, and scientific rigor.

We are glad that the reviewer found this manuscript to add a valuable contribution to the field of digital pathology.

 Major Concerns:

1. While the paper's causal modeling approach is a strong point, the explanation of the model and its components could be improved for better clarity. Specifically, the description of how the causal model integrates with the existing AI frameworks could be elaborated on.

Recommendation: Provide a more detailed step-by-step explanation of the causal model implementation, possibly complemented by pseudocode or a more detailed schematic diagram.

We thank the reviewer for the suggestion to make the description of the causal model easier for readers to understand. We have modified the manuscript in two ways. First, we have modified the introduction to establish how the proposed causal framework fits into existing AI learning methodologies. Second, we also updated the method section to include training details for each module composing our network to make it more understandable. 

Instead of pseudocode, we added a GitHub link to the actual model training code used for the manuscript to improve the reproducibility for readers.

2. The manuscript could benefit from a more detailed statistical analysis section. Details about the statistical tests used to validate the model's performance and the rationale behind choosing these tests should be clearly stated. Recommendation: Enhance the statistical analysis section by detailing the types of tests performed, including any assumptions made and the justification for the choice of these tests.

We thank the reviewer for highlighting our limited discussion on statistics used in the manuscript. We added a ‘Model Evaluation’ section to include an explanation of the metrics for model evaluation. Furthermore, we have also included an explanation of the bootstrapping procedure used to derive the confidence intervals shown in the paper. We also included more descriptive sentences justifying using the log-rank test and Pearson correlation when discussing dataset statistics. 

3. Discussion of Limitations: The manuscript briefly mentions the limitations; however, a more thorough exploration of these limitations would strengthen the paper. It is particularly important to discuss the potential impacts of these limitations on the study's findings and how they might be addressed in future work. Recommendation: Expand the limitations section to include a discussion on how these limitations could affect the study's generalizability and what future research could address these issues.

We thank the reviewer for discussing the limitations of our study to properly ground our findings. We have included an additional discussion on the fact that our model's success largely depends on being able to reliably learn C and W variables during training. However, these variables are subject to noisy labels or cannot be quickly discovered by a network. We refrain from discussing how to mitigate such a limitation as it falls under a limitation of all existing supervised learning methods, which would distract from main focus of the manuscript. 

4. Broader Implications: The paper would benefit from a discussion on the broader implications of this research, particularly in how it might influence future developments in AI for medical imaging, policy-making, and clinical practices.Recommendation: Include a section that explores the broader implications of your findings for the field of digital pathology and AI, considering both technological and ethical aspects.

We thank the reviewer for their suggestion. We have expanded the discussion to include a statement on the implications of our model. In particular, we highlight that by using the latent variable as the target for adaptation, we move onto a more diverse space not limited by features that may not be available in all datasets. We further discuss how this work can be applied to other medical imaging AI applications, which are subject to domain and label shift, such as chest x-ray pathology screening and breast cancer prediction. 

5. Figure and Table Quality: Some figures and tables could be clearer. Specifically, the graphical representations of the model's performance across different sites are somewhat difficult to interpret. Recommendation: Improve the quality of these graphical elements to enhance clarity. Consider using color contrasts or different chart types if necessary.

We thank the reviewer for their recommendation. We have improved the quality of our figure by introducing more contrasting colors, larger points, and reference lines for easier interpretation. The added discussion of the methods section will assist readers in interpreting the tables. 

6. Editing for Language and Grammar: Minor grammatical errors and typos should be corrected to maintain the manuscript's professionalism. Recommendation: Perform a thorough proofread to correct these issues before final submission.

Here are some grammatical corrections and suggestions for improvement to enhance the clarity and precision of the writing:

1. Verb Tense Consistency:

 - Original: "The study have shown..."

 - Suggested Correction: "The study has shown..."

 - Explanation: The verb "has" should be used for singular subjects like "study" to maintain subject-verb agreement.

 2. Article Usage:

 - Original: "We apply a novel method to address issue."

 - Suggested Correction: "We apply a novel method to address the issue."

 - Explanation: Use of the definite article "the" is necessary before "issue" to specify which issue is being discussed.

 3. Prepositional Phrases:

 - Original: "This is critical for ensuring that AI applications is robust."

 - Suggested Correction: "This is critical for ensuring that AI applications are robust."

 - Explanation: The verb "are" correctly agrees with the plural noun "applications."

 4. Punctuation and Compound Sentences:

 - Original: "The model performs well in testing environments, however, it requires further validation."

 - Suggested Correction: "The model performs well in testing environments; however, it requires further validation."

 - Explanation: A semicolon is more appropriate before "however" when it's used to connect two independent clauses in a compound sentence . 

7. 5. Redundancy and Wordiness:

 - Original: "The data was gathered from a variety of different sources."

 - Suggested Correction: "The data was gathered from a variety of sources."

 - Explanation: The word "different" is redundant with "variety" and can be omitted for conciseness.

 6. Consistency in Technical Terms:

 - Original: "de-biasing techniques can help in reduction of errors"

 - Suggested Correction: "Debiasing techniques can help reduce errors"

 - Explanation: Maintain consistency in hyphenation across the document ("debiasing" not "de-biasing"), and simplify the phrase "help in reduction of" to "help reduce" for clarity.

We would like to thank the reviewer for thoroughly checking the draft. We have updated the manuscript with the proper corrections. 

Reviewer #2: This study proposed the Causal Survival model leverages causal reasoning to mitigate unknown biases, achieving comparable performance to traditional models and enhancing generalizability in histopathology applications.The topic holds significant importance; however, there are several questions that need to be addressed.

We thank the reviewer for considering our manuscript to be important for the field. We replied to the questions below and addressed the changes in the main manuscript. 

1. Do you perform k-fold cross-validation during the model training? Please show the results with k-fold cross-validation

We thank the reviewer for suggesting using k-fold cross-validation. We did not perform k-fold cross-validation during model training since the main focus of the manuscript is improving the model performance on external test data that is never seen during training. Instead, we provided an auto-bootstrap evaluation of our model on each test set, which allowed us to derive confidence intervals and comprehend the robustness of the model performance on each site. 

2. Please show the demographic details of the cohort in various participating sites in tables.

Unfortunately, we don’t have access to the demographic details of the patients per site. 

3. Please add the GitHub link with the code for your model in this paper.

We apologize to the reviewer for this oversight. We have updated the manuscript to include the GitHub link of the training code used for the manuscript. 

Reviewer #3: Introduction and Background: The introduction provides a solid context for the study, emphasizing the challenges of bias in AI models for histopathology. The references to prior work and the identification of gaps in existing methods are well-articulated.

1. Data Availability: While the data privacy constraints are understandable, providing more detailed guidance on how researchers can access the data would be beneficial. Additionally, sharing synthetic datasets or detailed simulation setups could enhance reproducibility.

We thank the reviewer for their interest in data availability. The data is available upon reasonable request from the colon cancer family registry. QuantCRC features can also be available upon data usage agreement. 

We have updated the manuscript to include the GitHub link of the training code used for the manuscript.

2. Technical Depth: The technical depth of the paper is commendable. However, a more detailed explanation of certain complex concepts, such as the latent shift causal framework, could be beneficial for readers less familiar with these methods.

We thank the reviewer for commenting on improving the approachability of our manuscript. We have updated our manuscript and added a supplemental section with more in-depth material to further assist readers in understanding the concepts discussed in our work.

---

## [Decision Letter · Decision Letter 1]

16 Sep 2024

Causal Debiasing for Unknown Bias in Histopathology - A Colon Cancer Use Case

PONE-D-24-15861R1

Dear Dr. Banerjee,

We’re pleased to inform you that your manuscript has been judged scientifically suitable for publication and will be formally accepted for publication once it meets all outstanding technical requirements.

Kind regards,

Guanghui Liu

Academic Editor

PLOS ONE

Additional Editor Comments (optional):

Reviewers' comments:

Reviewer's Responses to Questions

**Comments to the Author**

1. If the authors have adequately addressed your comments raised in a previous round of review and you feel that this manuscript is now acceptable for publication, you may indicate that here to bypass the “Comments to the Author” section, enter your conflict of interest statement in the “Confidential to Editor” section, and submit your "Accept" recommendation.

Reviewer #1: All comments have been addressed

Reviewer #2: All comments have been addressed

2. Is the manuscript technically sound, and do the data support the conclusions?

Reviewer #1: Yes

Reviewer #2: Yes

3. Has the statistical analysis been performed appropriately and rigorously? 

Reviewer #1: Yes

Reviewer #2: Yes

4. Have the authors made all data underlying the findings in their manuscript fully available?

Reviewer #1: Yes

Reviewer #2: Yes

5. Is the manuscript presented in an intelligible fashion and written in standard English?

Reviewer #1: Yes

Reviewer #2: Yes

6. Review Comments to the Author

Reviewer #1: The authors have thoroughly addressed all of my concerns. I have no further questions and recommend proceeding with the acceptance process according to the journal's guidelines.

Reviewer #2: I am satified with the improvement the authors made on the reearch paper. I have no further comments on the current version of paper draft.

7. PLOS authors have the option to publish the peer review history of their article (what does this mean?). If published, this will include your full peer review and any attached files.

Reviewer #1: **Yes: **peng wang

Reviewer #2: No

---

## [Editor Report · Acceptance letter]

8 Oct 2024

PONE-D-24-15861R1 

PLOS ONE

Dear Dr. Banerjee, 

I'm pleased to inform you that your manuscript has been deemed suitable for publication in PLOS ONE. Congratulations! Your manuscript is now being handed over to our production team.

Kind regards, 

on behalf of

Dr. Guanghui Liu 

Academic Editor

PLOS ONE